# Association of a genetic risk score with BMI along the life-cycle: Evidence from several US cohorts

**Anna Sanz-de-Galdeano**[1,2,3]*, **Anastasia Terskaya**[4], **Angie Upegui**[1]

**1** FAE, University of Alicante, San Vicente del Raspeig, Alicante, Spain, **2** Institute of Labor Economics, Bonn, Germany, **3** CRES-UPF, Barcelona, Spain, **4** School of Economics and Business, University of Navarra, Pamplona, Spain

* anna.sanzdegaldeano@gmail.com

## Abstract

We use data from the National Longitudinal Study of Adolescent to Adult Health and from the Health and Retirement Study to explore how the effect of individuals' genetic predisposition to higher BMI —measured by BMI polygenic scores— changes over the life-cycle for several cohorts. We find that the effect of BMI polygenic scores on BMI increases significantly as teenagers transition into adulthood (using the Add Health cohort, born 1974-83). However, this is not the case for individuals aged 55+ who were born in earlier HRS cohorts (1931-53), whose life-cycle pattern of genetic influence on BMI is remarkably stable as they move into old-age.

## Introduction

According to the World Health Organization, worldwide obesity has almost tripled since 1975, and about 39% and 13% of the world's adult population in 2016 were overweight and obese, respectively. The prevalence of overweight and obesity among children and adolescents has risen even more dramatically from 4% in 1975 to just over 18% in 2016. The equivalent figures only for obesity among children and adolescents are just under 1% in 1975 and about 7% in 2016 (for further details see https://www.who.int/news-room/fact-sheets/detail/obesity-and-overweight).

These trends pose serious challenges to both individual and public health because raised BMI is a risk factor for noncommunicable conditions such as high cholesterol, high blood pressure, and coronary heart disease among others (see for instance [1] and the references therein), as well as some cancers [2] and mental illnesses [3, 4]. Additionally, obesity has also been shown to affect health care spending and individuals' socioeconomic outcomes (see for instance [5–9]).

Obesity is a many-sided problem with multiple determinants. Hence, its analysis has not been based on a unique perspective, and scholars from several disciplines have contributed to advance knowledge in this area. Social scientists have often focused on the role played by dietary and physical activity patterns that are in turn likely affected by factors like food prices,

**Data Availability Statement:** There are legal restrictions on sharing our data set publicly even if it is a de-identified/fully anonymized data set. Hereby you can see the requirements to access the restricted version of the data (which is the one we

use) established by the data producers: https://www.cpc.unc.edu/projects/addhealth/contracts; https://hrs.isr.umich.edu/data-products/access-to-public-data/conditions-of-use. If researchers are interested in using these data they must comply with those requirements. We hereby confirm that others would be able to access these data in the same manner we did, and we had no special privileges other would not have. Please submit your requests for data to: addhealth_contracts@unc.edu (for Add Health data) and hrsdapplication@umich.edu (for HRS data).

**Funding:** Anna Sanz-de-Galdeano acknowledges financial support from PROMETEO/2019/037 and from the Spanish Ministry of Economy and Competitiveness Grant ECO2017-87069-P.

**Competing interests:** The authors have declared that no competing interests exist.

agricultural policies, income, maternal employment and technology [8, 10, 11]. Importantly, BMI is also affected by genetic factors, and obesity is known to be both highly heritable and polygenic (see for instance [12–21]).

In this paper we study how the association between obesity-related genetic variants and BMI varies along the life-cycle or over time across several cohorts in the U.S., where obesity — which affects about 39% of adults— has increased dramatically in the past decades [22], and obesity-related conditions are some of the leading causes of preventable death [1]. We rely on data from two longitudinal representative surveys that contain genome-wide data from respondents: the National Longitudinal Study of Adolescent Health (Add Health hereafter) and the Health and Retirement Study (HRS hereafter). Individuals' genetic propensity to high BMI is measured using BMI polygenic scores —available in both Add Health and the HRS— constructed based on a recent large-scale genome wide association study for BMI [20]. We study whether the association between BMI and BMI polygenic scores is amplified or mitigated as teenagers transition and settle into adulthood (using Add Health), and as middle-age individuals transition to old-age (using the HRS). We also test whether significantly different patterns arise by childhood socioeconomic status and gender.

Our paper is related to a growing body of research that investigates how individuals' genetic predisposition to different phenotypes interacts with the environment [23, 24]. In regard to obesity, previous work has shown that childhood socioeconomic status [25], social understandings of body size [26], and individuals' education [27] moderate the influence of obesity-related genetic variants on obesity-related phenotypes.

Another related strand of the literature has instead used birth cohort as an indicator for exposure to obesogenic environment. Studies for the U.S. have shown that the association between obesity-related genetic variants and BMI is larger among individuals born in later cohorts [25, 28–30]. Additionally, [31] and [32] have uncovered an increase in the contribution of genetic factors to variation in BMI for successive birth cohorts in Sweden and Denmark, respectively. This body of results has been interpreted as evidence that individuals' genetic risk for elevated BMI is amplified when their lives unfold in more obesogenic socio-historical contexts.

This paper focuses on a related question that has received less attention in the literature: is the association between obesity-related genetic variants and obesity-related phenotypes attenuated or strengthened as individuals from the same cohort grow older? [33] have recently shown that the gap in the prevalence of severe obesity between individuals in the top and bottom polygenic score deciles widens during the transition from young adulthood to middle age in the U.S. (using data from the Framingham Offspring and Coronary Artery Risk Development in Young Adults studies), and they have also uncovered a similar pattern in children's weight from birth to 18 years of age in the UK (using data from the Avon Longitudinal Study of Parents and Children). [34] use the Dunedin Multidisciplinary Health and Development Study, which followed individuals born in 1972-73 in Dunedin (New Zealand) from birth through 38 years, and they find that higher BMI genetic risk scores predict higher BMI growth during childhood (from ages 3 through 13 years), as well as during adulthood (from ages 13 through 38 years).

We add to the limited literature on gene–age interaction effects on BMI [33–36] by analysing the effect of BMI polygenic scores as individuals transition from adolescence to young adulthood, and from middle-age to old-age. Moreover, we also analyse whether life-cycle profiles of genetic influence significantly differ by individual characteristics such as gender and socioeconomic status.

We find that the effect of BMI polygenic scores on BMI significantly increases as teenagers transition into adulthood. Specifically, our results for the younger cohort (Add Health, born

1974-84) indicate that a standard deviation increase in BMI polygenic scores is associated with a 4.2% increase in BMI at ages 15-16, while the percentage increase in BMI amounts to 5.7% when individuals are about 28. For the earlier HRS Original cohort (born 1931-41), the effect of BMI polygenic scores amounts to 4.2% when respondents are about 55 years old, and it remains stable while they transition into old-age and eventually reach age 72. We uncover similarly stable life-cycle patterns when focusing on subsequent HRS cohorts (born 1942-53) of 55 + individuals. Our main result is unchanged when analyzing individuals' life-cycle profiles separately by gender and socioeconomic status: the effect of BMI polygenic scores on BMI peaks in early adulthood.

Interestingly, we also find that, in our later Add Health cohort, the effect of BMI polygenic scores is significantly stronger for individuals with lower childhood socioeconomic status than for their higher socioeconomic status counterparts. In contrast, genetic influence on BMI does not significantly vary by socioeconomic status in any of the earlier HRS cohorts we analyze. We also find that the patterns of genetic influence on BMI do not significantly vary by gender, neither in Add Health nor in the HRS.

The remainder of the paper is organized as follows. The next section describes the data and methods used. The following two sections present respectively the results and discuss some robustness checks. The final section concludes.

## Materials and methods

We use data from Add Health and from the HRS. We now describe both datasets, as well as how our working samples have been constructed. We then explain the indicator we use to measure individuals' genetic predisposition to high BMI (which is available in both datasets), and outline our empirical model.

### The HRS dataset

The HRS is a nationally representative longitudinal study of the U.S. public over age 50 conducted every two years since 1992. The HRS collects information on health, socioeconomic background, employment, income, wealth, and other factors relevant to aging and retirement. Genotyping was performed using DNA samples collected during enhanced face-to-face interviews conducted on half of the HRS sample each wave starting in 2006 (and in later waves for new participants). Hence, respondents must have survived at least until genotyping started (2006-08) to be part of our analysis. Detailed information on the HRS genotype data and quality control process can be found at http://hrsonline.isr.umich.edu/modules/meta/xyear/pgs/desc/PGENSCORES3DD.pdf and http://hrsonline.isr.umich.edu/sitedocs/genetics/HRS2_qc_report_SEPT2013.pdf.

The HRS includes several birth cohorts with different entry years. In order to maximize sample size, our benchmark analysis is based on the so-called Original HRS cohort (born between 1931 and 1941) surveyed every two years from 1992 to 2012. However, we stop following this cohort in 2008 to avoid exacerbating potential biases related to mortality selection, which we discuss and address in Section Attrition. Our benchmark analysis relies on a balanced panel sample of 3,181 Original HRS cohort members of European descent who remained in the survey since 1992 until at least 2008, and for whom valid genetic data as well as information regarding their age, sex, height and weight are available. We focus on people of European descent because the BMI polygenic scores we use (described in detail in Section BMI Polygenic Scores) were constructed using the results of a genome-wide association study that mostly relied on a sample of European-descent individuals [20]. Based on self-reported height and weight information we have computed the Body Mass Index for respondents at

each wave using the standard formula: weight in kilograms divided by height in meters squared (kg/m$^2$). Individuals are classified as obese if their BMI is 30 or higher following the World Health Organization's recommendation regarding BMI thresholds for defining obesity in adults [37]. We use self-reports instead of measured values of weight and height because the latter are only available from 2006 onwards (see Section Objective Measurements versus Self-Reports of Weight and Height).

Table 1 provides basic descriptives on age, sex, BMI, and obesity prevalence for our analytic sample. Both mean BMI and obesity prevalence increase with age until individuals are almost 68, and they remain fairly stable thereafter at around 27.8 and 29%, respectively.

Additionally, we have replicated our analysis using two subsequent HRS cohorts: the War Babies cohort (born 1942-47 and followed from 1998 until 2014), and the Early Baby Boomers cohort (born 1948-55 and followed from 2004 until 2016). The sample selection criteria applied to these cohorts are analogous to those described above for the Original HRS cohort.

**Table 1. Summary statistics.** HRS Original Cohort Sample.

| Variable | Mean | Standard Deviation |
|---|---|---|
| BMI PGS (Normalized) | 0 | 1.000 |
| BMI 1992 | 26.951 | 4.562 |
| BMI 1994 | 27.055 | 4.555 |
| BMI 1996 | 27.267 | 4.739 |
| BMI 1998 | 27.536 | 4.838 |
| BMI 2000 | 27.689 | 4.935 |
| BMI 2002 | 27.779 | 4.936 |
| BMI 2004 | 27.871 | 5.126 |
| BMI 2006 | 27.909 | 5.191 |
| BMI 2008 | 27.825 | 5.247 |
| Obese 1992 | 0.217 | 0.412 |
| Obese 1994 | 0.235 | 0.424 |
| Obese 1996 | 0.242 | 0.429 |
| Obese 1998 | 0.263 | 0.440 |
| Obese 2000 | 0.277 | 0.448 |
| Obese 2002 | 0.289 | 0.453 |
| Obese 2004 | 0.296 | 0.456 |
| Obese 2006 | 0.292 | 0.455 |
| Obese 2008 | 0.290 | 0.454 |
| Age 1992 | 55.914 | 3.147 |
| Age 1994 | 57.776 | 3.140 |
| Age 1996 | 59.781 | 3.137 |
| Age 1998 | 61.649 | 3.135 |
| Age 2000 | 63.589 | 3.137 |
| Age 2002 | 65.733 | 3.137 |
| Age 2004 | 67.703 | 3.143 |
| Age 2006 | 69.677 | 3.130 |
| Age 2008 | 71.679 | 3.139 |
| Female | 0.553 | 0.497 |

Statistics based on a balanced panel sample of 3,181 HRS Original cohort members of European descent who remained in the survey from 1992 until at least 2008, and for whom valid genetic data as well as information regarding their age, sex, height and weight are available.

## The Add Health dataset

Add Health is a school-based longitudinal study of a nationally representative cohort of adolescents in grades 7-12 in the United States during the 1994-95 school year (n = 20,745, age range 12-20 at Wave 1). Add Health is based on a stratified sample of 80 high schools and 52 middle schools with probability of selection proportional to school size. Schools were stratified by region, urbanicity, school type, ethnic mix, and size. Add Health Wave I included an in-school questionnaire (administered to all the students attending the participating schools on the interview day), a more detailed in-home interview (conducted on a random sample of approximately 17 males and 17 females that were randomly selected within school and grade), and a parent questionnaire that was in general answered by the resident mothers of teenagers selected for the in-home sample. In-sample individuals have so far been followed in Waves II (1996, age range 12-21, n = 14,738), III (2000-01, age range 18-27, n = 15,197), IV (2008-09, age range 24-33, n = 15,701), and most recently in Wave V (2016-18, age range 33-43, n = 12,300).

We use data from all the waves of Add Health currently available (Waves I-V). Baseline demographic information on students and their families is obtained from Wave I, while self-reports on weight and height are used to construct BMI at each wave. We do not use objective measurements in our main analysis because they are not available in all waves of Add Health and the HRS. In Section Objective Measurements versus Self-Reports of Weight and Height, we replicate the main results using the objective BMI measures available in both datasets.

Saliva samples for DNA extraction were collected at Wave IV on the full sample. DNA measures were collected at Wave III for the sibling sample of Add Health (see https://www.cpc.unc.edu/projects/addhealth/documentation/guides/PGS_AH1_UserGuide.pdf for a detailed description of genome-wide data collection and quality control protocols).

The formula used to compute BMI is the same for children and adults ($kg/m^2$), but weight, height, and their relation to body fatness change along the life-cycle. All Wave I and most of Wave II respondents were still teenagers, so in those cases we followed the guidelines of the U.S. Centre for Disease Control and Prevention [38] and classified them as obese if their BMI was equal to or greater than the 95th percentile. BMI percentiles by sex and age in the US are taken from the 2000 CDC growth charts, publicly available at https://www.cdc.gov/growthcharts/percentile_data_files.htm. For respondents older than 20 we used instead the obesity definition applied to adults (BMI at or above 30).

Our Add Health analyses rely on a balanced panel sample of 2,730 individuals of European ancestry who remained in the survey from Wave I through Wave V with valid genetic data as well as information on age and sex, and for whom height and weight self-reports are available at all waves.

Table 2 provides basic descriptive statistics for this sample. There is a remarkable increase in both mean BMI (from 22.4 to 29.6) and obesity prevalence (which almost quadruples from 10% to 40%) as individuals transition from adolescence (average age 15.4) to young adulthood (average age 37.3).

## BMI polygenic scores

Both Add Health and the HRS currently include BMI polygenic scores, indices that summarize individuals' genetic risk for elevated BMI (*BMIPGS* hereafter). These BMI PGS were computed based on the genome wide association (GWAS) study for BMI conducted by [20] on a sample of 339,224 individuals. GWAS scan the entire genome in order to identify single nucleotide polymorphisms (SNPs) that are associated with a particular outcome while using strict significance thresholds to deal with multiple hypothesis testing. SNPs are locations in the genome

**Table 2. Summary statistics.** Add Health Sample.

| Variable | Mean | Standard Deviation |
|---|---|---|
| BMI PGS (Normalized) | -0.000 | 1.000 |
| BMI 1994/95 | 22.379 | 4.442 |
| BMI 1996 | 23.018 | 4.772 |
| BMI 2001/02 | 25.668 | 5.920 |
| BMI 2008/09 | 28.104 | 6.818 |
| BMI 2016/18 | 29.572 | 7.329 |
| Obese 1994/95 | 0.101 | 0.301 |
| Obese 1996 | 0.111 | 0.314 |
| Obese 2001/02 | 0.189 | 0.392 |
| Obese 2008/09 | 0.316 | 0.465 |
| Obese 2016/18 | 0.399 | 0.490 |
| Age 1994/95 | 15.412 | 1.702 |
| Age 1996 | 16.283 | 1.744 |
| Age 2001/02 | 21.738 | 1.747 |
| Age 2008/09 | 28.253 | 1.732 |
| Age 2016/18 | 37.305 | 1.839 |
| Female | 0.473 | 0.499 |

Statistics based on a balanced panel sample of 2,730 individuals of European ancestry who remained in the Add Health survey from Wave I through Wave V, and for whom valid genetic data as well as information regarding their age, sex, height and weight are available. Longitudinal weights are used.

where there are differences across individuals that can be associated with a particular trait. [39] provide further details regarding the construction of genetic risk scores from GWAS results. [20] used conservative thresholds for statistical significance ($P-value < 5 \times 10^{-8}$) and identified 97 SNPs significantly associated with BMI. *BMIPGS* are constructed for Add Health and HRS respondents by computing a weighted sum of these SNPs:

$$BMIPGS_i = \sum_{j=1}^{k} \hat{\beta}_j SNP_{ij} \qquad (1)$$

where $SNP_{ij} \in \{0, 1, 2\}$ is a count of the number of reference alleles for individual $i$ at SNP $j$, and $\hat{\beta}_j$ is the underlying GWAS coefficient estimated by [20] for each SNP associated with BMI. In our Add Health working sample, BMI polygenic scores account for 4.9% (Wave I in 1994-95, mean age 15.4), 5.5% (Wave II in 1996, mean age 16.3), 5.0% (Wave III in 2001-02, mean age 21.7), 6.2% (Wave IV in 2008-09, mean age 28.3) and 5.5% (Wave V in 2016-18, mean age 37.3) of the total variation in BMI. The corresponding figures for our HRS Original cohort analytic sample are: 6.2% (Wave I in 1992, mean age 55.9), 5.9% (Wave II in 1994, mean age 57.8), 6.3% (Wave III in 1996, mean age 59.8), 5.8% (Wave IV in 1998, mean age 61.6), 5.9% (Wave V in 2000, mean age 63.6), 5.8% (Wave VI in 2002, mean age 65.7), 5.9% (Wave VII in 2004, mean age 67.8), 5.5% (Wave VIII in 2004, mean age 69.7) and 5.2% (Wave IX in 2008, mean age 71.7).

Fig 1 plot the (kernel-smoothed) densities of respondents' *BMIPGS* in our HRS and Add Health balanced panel samples, respectively. The distributions are approximately normal.

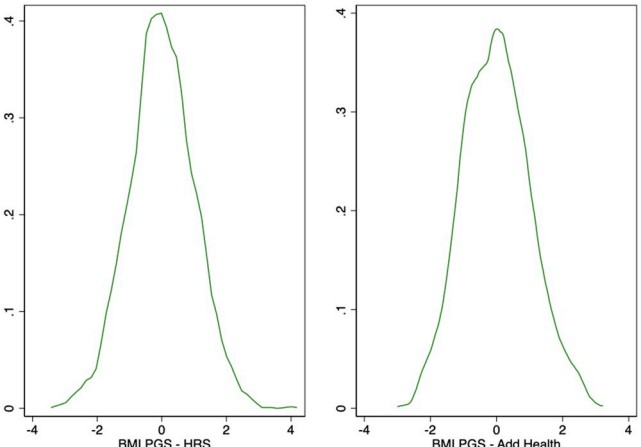

**Fig 1. BMI polygenic scores (Normalized) in the HRS original cohort and in add health. Kernel Density Estimates**. This figure displays the kernel-smoothed densities of HRS and Add Health respondents' BMI polygenic scores in the balanced samples described in Tables 1 and 2, respectively. Number of observations: 3,181 (HRS Original cohort) and 2,730 (Add Health).

## Empirical model

Our baseline empirical specification is:

$$Y_{ic,t} = \beta_0 + \beta_1 BMIPGS_{ic} + X'_{ic,t}\alpha + \epsilon_{ic,t}, \tag{2}$$

where $Y_{ic,t}$ is the log of BMI of individual $i$ observed at time $t$ who belongs to cohort $c$ (Add Health or the Original HRS cohort in our main analyses). $BMIPGS_{ic}$ denotes individuals' genetic predisposition to high BMI, which is fixed at conception. $BMIPGS_{ic}$ is standardized to have mean 0 and standard deviation 1. The vector $X_{ic,t}$ includes age, age squared, and a female dummy, as well as the 10 principal components of the full matrix of genetic data in order to account for population stratification [40, 41]. Our benchmark estimations of are based on self-reported BMI in order to avoid having different (objective vs. self-reported) BMI measurements for different ages. Add Health objective measures are available in Waves II-V (not in Wave I though, when individuals were 16.3 years old on average), while HRS objective measures are only available after 2006. In Section Objective Measurements versus Self-Reports of Weight and Height, we show estimation results based on objective BMI measures (whenever available), and compare them with our benchmark results based on subjective BMI measures. We estimate Eq (2) for the Add Health cohort (born 1974-83) and the Original HRS cohort (born 1931-41) at different points in time: 1994-95, 1996, 2001-02, 2008-09, and 2016-18 for Add Health, and every two years since 1992 until 2008 for the Original HRS cohort. We then analyze whether genetic influence on BMI is amplified or mitigated along the life-cycle for both Add Health respondents (as they transition from adolescence to young adulthood) and Original HRS cohort members (as they transition from middle-age to old-age). Our choice of a log-level model rather than a level-level model in Eq (2) is supported by AIC test results. In line with this, unconditional regression estimates [42] indicate (see S1 Appendix of Fig 1 and 2) that the effect of a standard deviation increase in *BMIPGS* on BMI is non-linear and it is larger (in absolute terms) the higher the level of BMI.

## Main results

### Genetic influence on BMI along the life-cycle: General patterns

The results of estimating Eq (2) on the sample of HRS Original cohort members are summarized in Fig 2, which depicts OLS coefficient estimates of $\beta_1$ (as well as their associated 95% confidence intervals) that measure the estimated percentage increases in BMI associated with a standard deviation increase in *BMIPGS* as middle-aged adults move into old-age. OLS coefficient estimates and their corresponding standard errors (clustered at the household level) are displayed in S1 Appendix of Table 1. The estimated life-cycle profile indicates that BMI increases associated with a standard deviation increase in *BMIPGS* remain stable around just above 4% along the life-cycle. Interestingly, a similarly flat life-cycle profile is observed in two subsequent HRS cohorts—the HRS War Babies cohort (born 1942-47) and the Early Baby Boomers cohort (born 1948-1953)—for whom a standard deviation increase in *BMIPGS* is associated with BMI increases of 5%-6% as they grow older (see Fig 3 in S1 Appendix).

Results for Add Health respondents (Fig 3, S1 Appendix of Table 2) instead suggest that the association between *BMIPGS* and BMI increases as teenagers become adults. In particular, a standard deviation increase in *BMIPGS* increases individuals' BMI by 5.8% by the time they are about 37 in 2016-18, a significantly larger association than the one estimated (4.2%) when they were 15-16 years old (in 1994-95, one sided p-value = 0.002). Interestingly, the association between *BMIPGS* and log(BMI) appears to stabilize at just above 5.5% at Wave IV (2008-09, average age 28).

Eq (2) is a reduced-form model, and disentangling the mechanisms through which individuals' *BMIPGS* may differently affect their BMI at different stages of their lives is beyond the scope of this paper. However, it is worth outlining several potential (and not mutually exclusive) determinants of the pattern of genetic influence we uncover in Add Health. First, homophily may be playing a role both at the genotypic [43] and the phenotypic level [44–46]. In the presence of peer effects, homophily may, in turn, lead to social multiplier effects, which would be consistent with the increasing relevance of genetic influence we observe until Add Health individuals approach age 30. These effects may, however, dissipate over time. For instance,

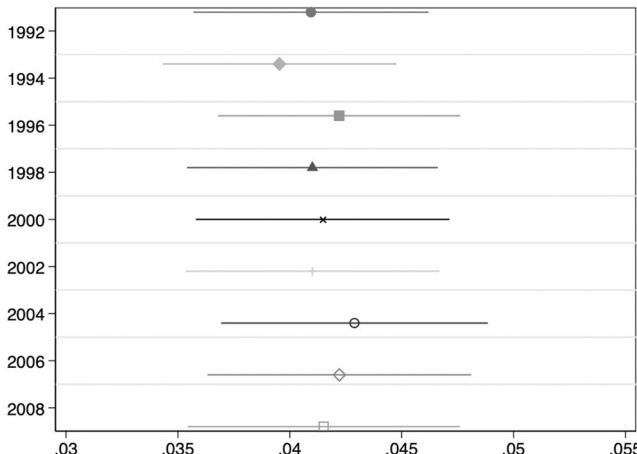

**Fig 2. Association between BMI polygenic scores and Log(BMI) along the life-cycle. HRS Original Cohort**. This Figure summarizes the results of estimating Eq 2 on the balanced sample of 3,181 HRS Original cohort members described in Table 1. The dependent variable is Log(BMI). OLS coefficient estimates of $\beta_1$ as well as their associated 95% confidence intervals are depicted. All regressions include a female dummy, age, age squared, and the first 10 principal components of the full matrix of genetic data. Standard errors are clustered at the household level.

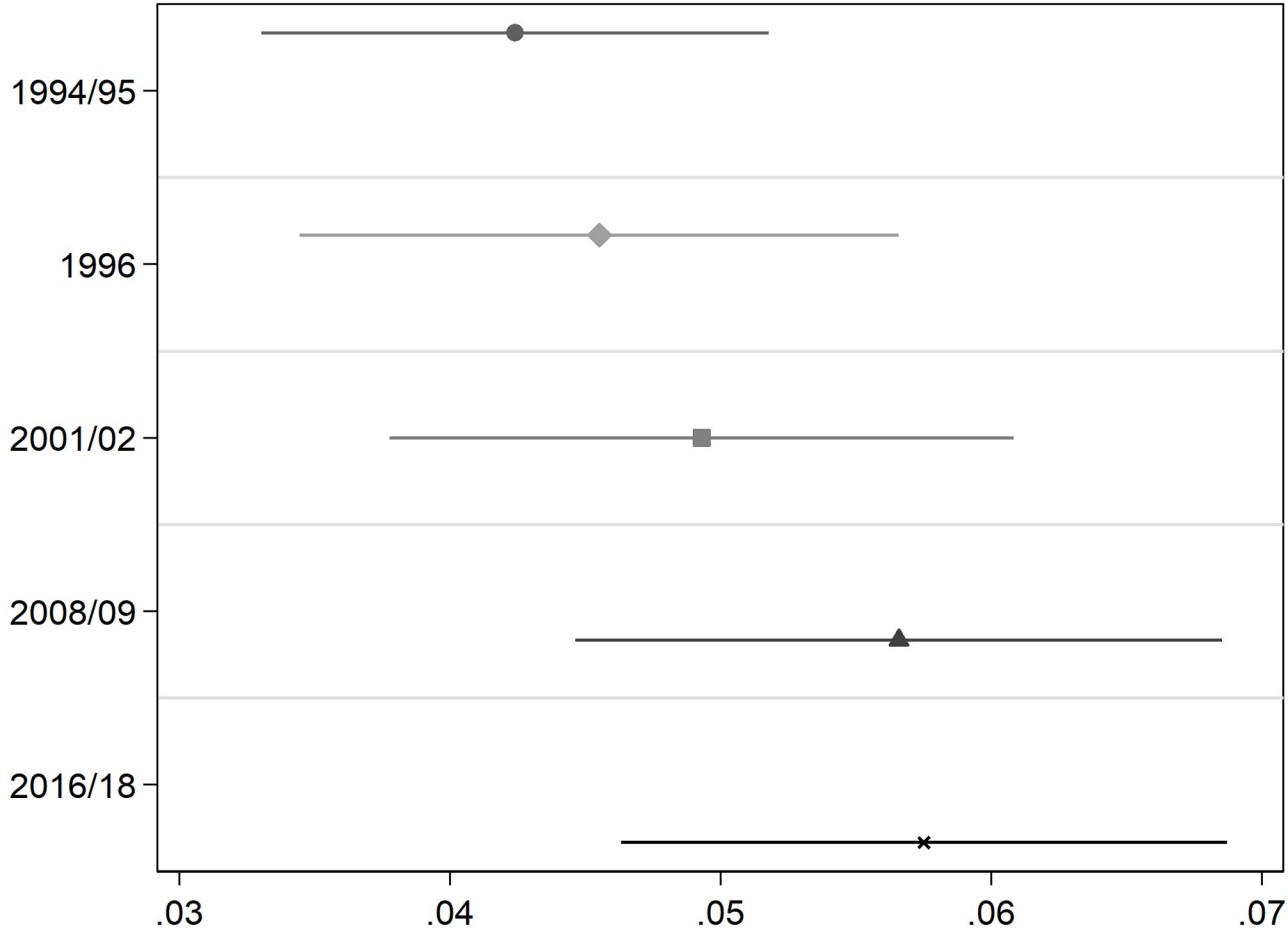

**Fig 3. Association between BMI polygenic scores and Log(BMI) along the life-cycle. Add Health Cohort.** This Figure summarizes the results of estimating Eq 2 on the balanced sample of 2,730 Add Health cohort members described in Table 2. The dependent variable is Log(BMI). OLS coefficient estimates of $\beta_1$ as well as their associated 95% confidence intervals are depicted. All regressions include a female dummy, age, age squared, and the first 10 principal components of the full matrix of genetic data. Standard errors are clustered at the school level. Longitudinal weights are used.

[47] find that social-genetic effects on obesity induced by interactions with high school grade-mates are relevant for girls in adolescence, but they dissipate as they grow into adulthood.

Second, the effect of genes on BMI is likely to be altered by environmental factors that change during the life course [25–27, 48]. For example, [48] suggests that individuals with high genetic predisposition for obesity are more responsive to food intake than those with low genetic predisposition for obesity. To the extent that the impact of food consumption on BMI accumulates over time, the BMI gap between individuals with low and high *BMIPGS* can also grow throughout life.

Third, individuals with high *BMIPGS* may sort into more obesogenic environments. In line with this hypothesis, [48] shows that individuals with a higher genetic predisposition for obesity tend to display a higher demand for food, the effect of which can also be cumulative. In contrast, [49] find that higher *BMIPGS* are associated with weight loss behaviors, which could reduce genetic influence throughout life.

Fourth, genetically influenced characteristics in children may evoke environmental responses that may in turn alter those characteristics, as genetic and environmental variation are not mutually exclusive [50].

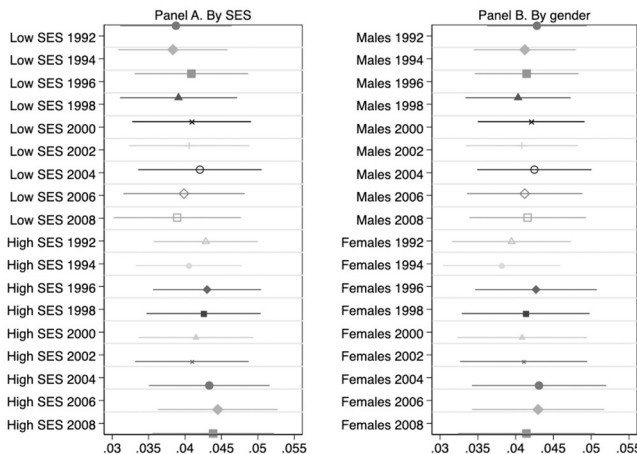

**Fig 4. Association between BMI polygenic scores and Log(BMI) along the life-cycle: Patterns by gender and socioeconomic background. HRS Original Cohort**. This Figure summarizes the results of estimating Eq 2 on the balanced sample of 3,181 HRS Original cohort members described in Table 1 by parental socioeconomic status (SES, in Panel A) and by gender (Panel B). The dependent variable is Log(BMI). OLS coefficient estimates of $\beta_1$ as well as their associated 95% confidence intervals are depicted. All regressions include age, age squared, and the first 10 principal components of the full matrix of genetic data. Regressions by SES (in Panel A) also include a female dummy as a covariate. Low and High SES individuals are those whose parental socioeconomic status is below and above the median, respectively. Standard errors are clustered at the household level.

Finally, there may be age-related differences in genetic expression which may result in later manifestations of some genes [51, 52].

## Genetic influence on BMI along the life-cycle: Patterns by gender and socioeconomic background

We now explore whether the life-cycle patterns we have uncovered so far significantly vary by gender and by childhood socioeconomic status (SES). We use parental background information from both Add Health and the HRS in order to construct summary indices of childhood SES. The construction of these summary indices is detailed in S1 Appendix. Individuals are classified as High SES and Low SES if the value of their childhood SES index is above and below the median, respectively.

Fig 4 shows how genetic influence on BMI varies by gender and by socio-economic status in the sample of HRS Original cohort members as they age. The life-cycle profile of genetic influence is stable for all subgroups. Additionally, the association between *BMIPGS* and log (BMI) does not significantly differ neither by gender nor by SES at any point in time.

The results for the Add Health cohort are depicted in Fig 5. There is a remarkable SES gradient in the influence of *BMIPGS*: the effect of BMI polygenic scores is significantly stronger for individuals with lower family socioeconomic status than for those with higher socioeconomic status. In contrast, there are no significant differences by gender.

Regarding life-cycle patterns of genetic influence in Add Health, the conclusions are the same for all subgroups: the association between *BMIPGS* and log(BMI) significantly increases as adolescents transition into adulthood.

## Additional results

### Pubertal stage and the association of BMI PGS with BMI

Puberty and BMI are likely related ([53, 54], among others), and pubertal timing differs across individuals. Therefore, part of BMI variation during adolescence may be due to pubertal stage

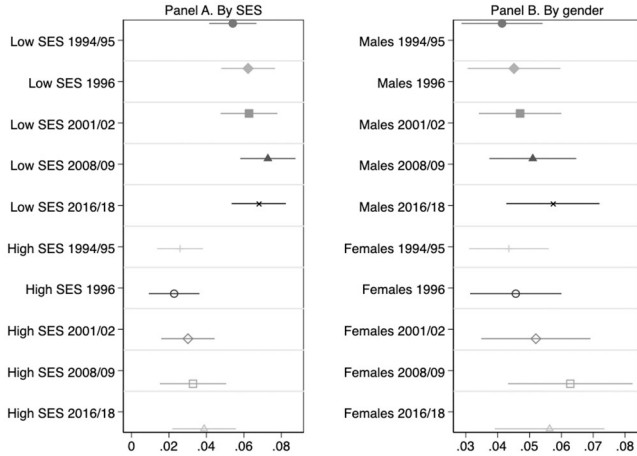

**Fig 5. Association between BMI polygenic scores and Log(BMI) along the life-cycle: Patterns by gender and socioeconomic background. Add Health Cohort**. This Figure summarizes the results of estimating Eq 2 on the balanced sample of 2,730 Add Health cohort members described in Table 2 by parental socioeconomic status (SES, in Panel A) and by gender (Panel B). The dependent variable is Log(BMI). OLS coefficient estimates of $\beta_1$ as well as their associated 95% confidence intervals are depicted. All regressions include age, age squared, and the first 10 principal components of the full matrix of genetic data. Regressions by SES (in Panel A) also include a female dummy as a covariate. Low and High SES individuals are those whose parental socioeconomic status is below and above the median, respectively. Standard errors are clustered at the school level. Longitudinal weights are used.

differences across teenage respondents. Hence, the variance of the error in Eq (2) is likely larger for adolescents than for older individuals. Moreover, there is evidence that pubertal timing and BMI have a common genetic component and therefore part of the effect of genes on BMI might be explained by the effect of genes on pubertal timing [55, 56]. To study whether our previous results are affected by these factors, we replicate our baseline analyses including gender-specific information on the stage of development of adolescents that Add Health collected in Waves I and II, as by Wave III individuals were already between 18 and 26 years old (21.7 years old on average in our analytic sample).

In particular, we use the following questions that were asked to boys in Waves I and II: i) "How much hair is under your arms now? 1 I have no hair at all, 2 I have a little hair, 3 I have some hair, but not a lot; it has spread out since it first started, 4 I have a lot of hair that is thick, 5 I have a whole lot of hair that is very thick, as much hair as a grown man"; ii) "How thick is the hair on your face? 1 I have a few scattered hairs, but the growth is not thick, 2 The hair is somewhat thick, but you can still see a lot of skin under it, 3 The hair is thick; you can't see much skin under it, 4 The hair is very thick, like a grown man's facial hair"; iii) "Is your voice lower now than it was when you were in grade school? 1 No, it is about the same as when you were in grade school, 2 Yes, it is a little lower than when you were in grade school, 3 Yes, it is somewhat lower than when you were in grade school, 4 Yes, it is a lot lower than when you were in grade school, 5 Yes, it is a whole lot lower than when you were in grade school; it is as low as an adult man's voice"; and iv) "How advanced is your physical development compared to other boys your age? 1 I look younger than most, 2 I look younger than some, 3 I look about average, 4 I look older than some, 5 I look older than most".

As for girls, we use the following questions that were asked in Waves I and II: i) "As a girl grows up her breasts develop and get bigger. Which sentence best describes you? 1 My breasts are about the same size as when I was in grade school, 2 My breasts are a little bigger than when I was in grade school, 3 My breasts are somewhat bigger than when I was in grade school, 4 My breasts are a lot bigger than when I was in grade school, 5 My breasts are a whole

lot bigger than when I was in grade school, they are as developed as a grown woman's breasts";
ii) "As a girl grows up her body becomes more curved. Which sentence best describes you? 1
My body is about as curvy as when I was in grade school, 2 My body is a little more curvy than
when I was in grade school, 3 My body is somewhat more curvy than when I was in grade
school, 4 My body is a lot more curvy than when I was in grade school, 5 My body is a whole
lot more curvy than when I was in grade school"; iii) "Have you ever had a menstrual period
(menstruated)? 0 No, 1 Yes"; and iv) "How advanced is your physical development compared
to other girls your age? 1 I look younger than most, 2 I look younger than some, 3 I look about
average, 4 I look older than some, 5 I look older than most". We construct binary indicators
for all the possible answers to these questions and we add them as controls to our estimations
of Eq (2) for Waves I and II. The results of this analysis, reported in S1 Appendix of Table 3,
indicate that the effect of *BMIPGS* on log(BMI) is lower after the inclusion of puberty stage
controls. This is consistent with the fact that pubertal timing and BMI have a common genetic
component. As a consequence, the estimated association between *BMIPGS* and log(BMI)
increases more markedly as individuals transition from adolescence into adulthood when we
control for pubertal stage indicators than when we do not (see Fig 3 and/or S1 Appendix of
Table 2). While it is reassuring that our conclusion is robust to the addition of pubertal stage
indicators, our preferred specification excludes this set of controls in order to avoid reverse
causality bias, as there is evidence that childhood obesity increases the risk of premature
puberty for girls and boys [54]. Moreover, we re-estimate our benchmark model including
pubertal timing as an additional regressor in S1 Appendix of Table 4. Females' puberty onset is
classified as early vs. delayed if age of menarche was lower 13 (which is the median in our sam-
ple) vs. 13+. Establishing males' puberty onset is more complex. We do so following the rec-
ommendations from [57]. In particular, we regress a pubertal status index on age, and we then
save the residuals. The pubertal status index has been constructed using principal component
analysis on the variables related to pubertal stage for boys previously described and measured
in Wave I, as they display more variation in Wave I than in Wave II. Males' puberty onset is
subsequently classified as early vs. delayed if the regression's residuals are below vs. above the
median. As the comparison between Columns 1 and 2 of S1 Appendix Table 4 reveals, the
inclusion of pubertal timing as a control barely alters the estimated coefficients of *BMIPGS*. In
summary, this evidence indicates that the increasing pattern of association between *BMIPGS*
and log(BMI) we find for Add Health adolescents as they transition into adulthood is robust to
the inclusion of controls for pubertal stage and the timing of puberty onset.

## Morbidity and the association of BMI PGS with BMI

Chronic diseases are more prevalent among the elderly, and they may in turn lead to wasting
(BMI loss). We investigate whether our previous results for HRS Original cohort members are
affected by the prevalence of the following conditions: heart disease, cancer, diabetes, lung dis-
ease, and arthritis. First, we study how the prevalence of these conditions correlates with both
BMI and with *BMIPGS* in our analytic sample. The prevalence of heart disease, diabetes, and
arthritis is positive and significantly correlated with BMI, while the prevalence of cancer, lung
disease, and BMI are not significantly correlated. This pattern is the same for all sample years,
that is, since individuals are on average 55.9 years old (in 1992) until they reach 71.7 years of
age on average (in 2008). Hence, we find no evidence of BMI reductions being linked to higher
prevalence of chronic diseases in our sample. The correlation between *BMIPGS* and chronic
diseases is positive and significant for heart disease, diabetes, and arthritis, while it is generally
insignificant for cancer and lung disease.

Next, we replicate our baseline analyses including the prevalence of these five chronic conditions as additional controls in all our sample years. The results of this analysis, reported in S1 Appendix Table 5 reveal that the inclusion of this set of controls slightly attenuates the estimated association between *BMIPGS* and log(BMI). This is consistent with our previous finding that *BMIPGS* are positively and significantly correlated with several chronic diseases. Importantly, the life-cycle association between *BMIPGS* and log(BMI) remains stable as individuals transition from middle-age to old-age once these additional controls are included in our benchmark model (2). However, we do not include them in our preferred specification because their relationship with BMI is likely bidirectional.

## Robustness checks

### Attrition

The longitudinal nature of our analyses implies that there is attrition in both our Add Health and HRS samples. This could be problematic if attrition is systematically related to *BMIPGS*. We cannot directly test whether this is the case because individuals were not genotyped in the first wave we observe them neither in Add Health (genotyping took place in Wave IV) nor in the HRS (genotyping took place in 2006-08).

We can, however, investigate whether attrition is related to obesity and BMI measured the first time individuals were interviewed. We do so by regressing a binary variable identifying missing individuals due to attrition between the first and the last waves analyzed on initial BMI and obesity. We find that attrition is not significantly related to initial BMI or obesity status neither in Add Health nor in the HRS.

Concerns about attrition due to selective mortality may remain in the HRS because members of the HRS Original cohort (55.9 years old on average the first time we observe them in 1992) may have died by the time genetic data were collected [58, 59], and elevated BMI is known to have adverse health consequences. Actually, if we regress a dummy identifying attrited individuals due to death between the first (1992) and the last wave (2012) analyzed (instead of a dummy identifying overall attrition) on BMI and obesity measured in 1992, the estimated coefficients are positive and significant. Hence, we adjust our benchmark results for the HRS Original cohort by using inverse probability weighting methods as in [59]. Fitted values from a logit survival regression are used to obtain probability weights which are used as inverse probability weights to adjust estimates for selective mortality. In particular, our inverse probability weights are based on fitted values obtained from estimating a logit model of the probability of survival (until genotyping took place) as a function of respondents' educational attainment, year of birth, and several health indicators (the means of individuals' BMI, CES depression scale, and self-reported health over all available years, indicators of whether respondents ever reported smoking, having diabetes, and having heart disease, and respondents' maximum height over all available waves).

The results of this adjustment, presented in Table 6 in S1 Appendix, suggest that our results are robust to selective mortality because they are extremely similar to those obtained in our benchmark analysis.

### Objective measurements versus self-reports of weight and height

Objective measurements of height and weight are only available in some waves of Add Health (Waves II, III, IV, and V) and the HRS (2006 and 2008). We use this information to investigate whether it is likely that using self-reports may affect our results, and our findings are reassuring. We show estimation results based on objective BMI measures (whenever available), and compare them with our benchmark results based on subjective BMI measures in Table 7 in

S1 Appendix. Panel A of Table 7 in S1 Appendix displays the estimated associations between *BMIPGS* and objective (Column 1) and self-reported (Column 2) log(BMI) for the HRS Original cohort for years 2006 and 2008 (our sample years with available objective BMI measures). The comparison of Columns 1 and 2 reveals that the estimated associations between *BMIPGS* and objective and self-reported log(BMI) barely differ. Therefore, our conclusion that the link between *BMIPGS* and log(BMI) is stable over as middle-age individuals transition to old-age remains when using objective BMI measures. Panel B of Table 7 in S1 Appendix does the same comparative analysis for the Add Health cohort. The estimated coefficients of *BMIPGS* do not significantly differ (at the 5% level) across columns for all waves. Importantly, our finding that the association between *BMIPGS* and log(BMI) increases as adolescents transition into adulthood prevails when using objective BMI measures.

## Socioeconomic status and the association of BMI PGS with BMI

We now replicate our benchmark analyses including childhood SES among the set of control variables. This allows us to explore further whether the observed life-cycle associations between *BMIPGS* and log(BMI) reflect similar patterns as association between SES and log(BMI) as individuals grow older. The results of these analyses are shown in Tables 8 and 9 in S1 Appendix.

The association between SES and log(BMI) for Add Health cohort members is negative, significant, and it increases (in absolute terms) as they transition from adolescence into adulthood (Table 8 in S1 Appendix, Column 2). However, the inclusion of SES among the control set barely changes the estimated coefficients *BMIPGS* (Table 8 in S1 Appendix, comparison of Columns 1 and 3). This indicates that SES effects across the life course cannot explain the observed increasing association between *BMIPGS* and log(BMI) between adolescence and early adulthood, which remains basically unaltered when SES is held constant.

The association between SES and log(BMI) for HRS Original cohort members is negative and significant, and it does not significantly change as individuals get older (Table 9 in S1 Appendix, Column 2). The inclusion of the childhood SES index among the set of control variables hardly modifies the estimated coefficients of *BMIPGS* (Table 9 in S1 Appendix, comparison of Columns 1 and 3).

In summary, this evidence indicates that SES cannot account for the life-cycle patterns of association between *BMIPGS* and log(BMI) we have uncovered so far, neither for Add Health nor for HRS Original cohort members.

## Discussion

In this paper we find that the effect of BMI polygenic scores on log(BMI) increases significantly as teenagers transition into adulthood (using the Add Health cohort, born 1974-83). However, this is not the case for individuals aged 55+ who were born in earlier cohorts (HRS Original cohort born 1931-41, War Babies cohort born 1942-47, and Early Baby Boomers cohort born 1948-53), whose life-cycle pattern of genetic influence on BMI is remarkably stable. We uncover similar life-cycle patterns for all the cohorts we study when we separately analyse males and females, and low and high socioeconomic status groups.

One possible explanation for our results is that the effect of BMI polygenic scores on BMI increases until people reach a certain age, and remains stable thereafter. This hypothesis is consistent with [36], who find that the association between genes and BMI peaks in early adulthood.

Interestingly, we also find that the association between BMI polygenic scores and BMI significantly differs by childhood socioeconomic status in the Add Health cohort, while this is

not the case in earlier HRS cohorts. In particular, childhood socioeconomic status significantly moderates the effect of BMI polygenic scores for Add Health cohort members. In contrast, the effect of BMI polygenic scores does not significantly differ by gender in any of the cohorts analysed.

Last but not least, our findings also indicate that the effect of BMI polygenic scores on BMI is likely to be non-linear. In fact, the AIC test rejects the linear model in favor of a log-linear model. This simple transformation might be considered when conducting future GWAS in order to improve the predictive power of polygenic scores.

## Strengths and limitations

In this paper, we use two longitudinal surveys to provide new evidence on gene-age interaction effects on BMI for several cohorts. In particular, we study teenagers from the Add Health cohort (born 1974-1983) as they transition into adulthood as well as individuals aged 55+ who were born in earlier HRS cohorts (1931-53) as they move into old-age. The use of panel data is crucial in this context because it allows one to disentangle age/time associations from cohort effects. In contrast, as argued by [35], cross-sectional studies may fail to detect age-varying associations as they cannot disentangle age/time from cohort effects. Our analyses are based on different cohorts observed at different stages of the life cycle. Hence, our contrasting findings for Add Health and the HRS may reflect differing patterns of genetic influence along the life cycle, but they could also stem from systematic differences across cohorts in their life-cycle patterns of genetic influence.

Note also that in this paper we estimate a reduced-form model without digging into the mechanisms behind gene-age interactions because of data limitations. Our results therefore could be explained by changes in the biology of BMI across the life course as well as by environmental changes that may reinforce or mitigate the effect of genes on BMI [25, 27, 60]. Understanding the mechanisms behind the patterns we uncover is worth further investigation.

Another limitation of our analyses is that the genome-wide association study employed to compute the *BMIPGS* used mostly relies on European-descent individuals [20]. Therefore, our results cannot be generalized to individuals from different ancestries. The availability of GWAS for other ancestries would allow to overcome this limitation.

Finally, another potential limitation stems from the fact that the strength of genotype-phenotype associations may vary by age. Hence, GWAS results may not replicate in samples where the age distribution differs from that of the GWAS sample [35]. The *BMIPGS* we use rely on the GWAS conducted by [20], which is in turn mostly based on a sample of midlife individuals. Hence, their predictive power may be lower for younger individuals. A similar argument may apply to other demographic characteristics like childhood socioeconomic status (as our Add Health results by socioeconomic status suggest). While the strongest *BMIPGS-BMI* association we uncover is for young adults (Waves 4 and 5 of Add Health), this warrants further investigation.

## Supporting information

**S1 Appendix. Additional figures and tables and childhood socioeconomic status indices construction.**
(PDF)

**S1 File.**
(ZIP)

## Acknowledgments

We thank Pedro Albarrán, Dimitris Christelis, and Michael Rosholm for comments and suggestions.

## Author Contributions

**Conceptualization:** Anna Sanz-de-Galdeano, Anastasia Terskaya, Angie Upegui.

**Data curation:** Anna Sanz-de-Galdeano, Anastasia Terskaya, Angie Upegui.

**Formal analysis:** Anna Sanz-de-Galdeano, Anastasia Terskaya, Angie Upegui.

**Investigation:** Anna Sanz-de-Galdeano, Anastasia Terskaya, Angie Upegui.

**Methodology:** Anna Sanz-de-Galdeano, Anastasia Terskaya, Angie Upegui.

**Writing – original draft:** Anna Sanz-de-Galdeano, Anastasia Terskaya, Angie Upegui.

**Writing – review & editing:** Anna Sanz-de-Galdeano, Anastasia Terskaya, Angie Upegui.

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
