## [Decision Letter · Decision Letter 0]

10 Mar 2020

PONE-D-20-03465

Association of a Genetic Risk Score with BMI along the Life-Cycle: Evidence from several US Cohorts

PLOS ONE

Dear Dr Sanz-de-Galdeano

Thank you for submitting your manuscript to PLOS ONE. After careful consideration, we feel that it has merit but does not fully meet PLOS ONE’s publication criteria as it currently stands. Therefore, we invite you to submit a revised version of the manuscript that addresses the points raised during the review process.

We would appreciate receiving your revised manuscript by Apr 24 2020 11:59PM. To enhance the reproducibility of your results, we recommend that if applicable you deposit your laboratory protocols in protocols.io, where a protocol can be assigned its own identifier (DOI) such that it can be cited independently in the future. For instructions see: http://journals.plos.org/plosone/s/submission-guidelines#loc-laboratory-protocols

We look forward to receiving your revised manuscript.

Kind regards,

David Meyre

Academic Editor

PLOS ONE

Journal Requirements:

2) In the ethics statement in the manuscript and in the online submission form, please provide additional information about the patient records used in your retrospective study. Specifically, please ensure that you have discussed whether all data were fully anonymized before you accessed them and/or whether the IRB or ethics committee waived the requirement for informed consent. If patients provided informed written consent to have data from their medical records used in research, please include this information.

3) In your Methods section, please provide additional information about the participant inclusions method and the demographic details of your participants. Please ensure you have provided sufficient details to replicate the analyses such as: a)  a description of any inclusion/exclusion criteria that were applied to participant inclusion, b) a table of relevant demographic details, c) a statement as to whether your sample can be considered representative of a larger population.

4) We note that you have indicated that data from this study are available upon request. PLOS only allows data to be available upon request if there are legal or ethical restrictions on sharing data publicly. For information on unacceptable data access restrictions, please see http://journals.plos.org/plosone/s/data-availability#loc-unacceptable-data-access-restrictions.

5)  Thank you for stating the following in the Acknowledgments/(Title page) Section of your manuscript:

[She acknowledges financial support from PROMETEO/2019/037, and

from the Spanish Ministry of Economy and Competitiveness Grant ECO2017-87069-P]

 [The author(s) received no specific funding for this work.]

Please include the updated Funding Statement in your cover letter. We will change the online submission form on your behalf.

Reviewers' comments:

Reviewer's Responses to Questions

**Comments to the Author**

1. Is the manuscript technically sound, and do the data support the conclusions?

Reviewer #1: Partly

Reviewer #2: Yes

2. Has the statistical analysis been performed appropriately and rigorously? 

Reviewer #1: Yes

Reviewer #2: Yes

3. Have the authors made all data underlying the findings in their manuscript fully available?

Reviewer #1: Yes

Reviewer #2: Yes

4. Is the manuscript presented in an intelligible fashion and written in standard English?

Reviewer #1: Yes

Reviewer #2: Yes

5. Review Comments to the Author

Reviewer #1: This article reports analysis of genetic association with BMI across the life course. The authors ask if genetic associations with BMI are stronger at some ages as compared to others. They use data from two large social surveys that include longitudinal assessments of BMI. The main finding is that genetic associations with BMI are somewhat stronger at older ages up to the 50s/60s, when associations stabilize.

These findings are consistent with previous reports in both the twin and molecular genetic literatures. The authors should note the Belsky et al. 2012 paper, which was the first to report polygenic score associations with BMI development across the first half of the life course. (the Hardy et al. paper cited analyzed only the FTO polymorphism)

The main technical criticism I would make of this analysis is that the authors do not take seriously enough the limitation that much of their data come from self-reports of height and weight. They tell us they did some analysis to evaluate the risk this posed to the inferences they make but don’t show the data. They should.

This is particularly important for interpreting genetic associations with BMI across the transition from adolescence to young adulthood. The adolescent BMI data come from self-reports whereas the young adult data come from anthropometric assessments. If there is less error in measurement in the anthropometry-derived BMIs, this will lead to larger effect-sizes in association analysis.

A second critique is that the authors don’t seem to think much about the biology of BMI change across the life course and how this may affect genetic associations. Two processes are of particular relevance to the analysis reported. First, puberty causes substantial changes in BMI. Pubertal timing varies across individuals. Variation in pubertal timing may therefore result in a kind of measurement error in the BMI phenotype being analyzed in adolescence, biasing genetic effect-sizes toward the null. Second, with advancing age, a range of chronic diseases become more prevalent, leading to wasting (BMI loss). Add Health data on timing of menarche and HRS data on chronic disease morbidity may be helpful in exploring these processes.

Basically, my concern is that the authors are not identifying substantive differences in how genetics affect BMI, but instead are observing variation in the magnitudes of non-genetic causes (or genetic causes not measured in the PGS) across life course stages.

One idea to explore in evaluating this issue is to use some non-genetic measure of risk for obesity. For example, both Add Health and HRS measure parental education, which is associated with BMI across the life course. Do parental education associations with BMI show the same patterns of change with age as genetic associations? If so, is this analysis telling us something about genetics or simply about the sources of systematic variation in BMI? If not, this is a strong piece of evidence that the patterning observed is specifically about the genetics being studied.

MINOR

To my mind, there is a conceptual problem with the article. The authors approach their question within a GxE framework. But the changing association of genetics with BMI as people age is not a GxE. BMI growth is a developmental process, with BMI at later ages strongly influenced by BMI earlier on. Given the authors have repeated measures data on individuals, the question they should be asking is how do BMI genetics influence BMI change across the life course. But this is a matter of taste and differing views of how this problem should be approached ought not to interfere with publication in this journal.

Something else: The polygenic score analyzed by the authors comes from a GWAS that included mainly midlife individuals. For this reason, we might expect the strongest genetic associations in that age range. This might be discussed somewhat more in the introduction and discussion sections of the article.

Reviewer #2: This is an excellent study, well performed and well written, and I recommend its publication.

Statistical methods are explained in detail and results are clear.

There is a typo in the abstract (a duplicated "the").

Congratulations.

6. PLOS authors have the option to publish the peer review history of their article (what does this mean?). If published, this will include your full peer review and any attached files.

Reviewer #1: No

Reviewer #2: No

---

## [Author Response · Author response to Decision Letter 0]

19 Jun 2020

See attached letter named Response to Reviewers.docx

---

## [Editor Report · Decision Letter 1]

20 Jul 2020

PONE-D-20-03465R1

Association of a Genetic Risk Score with BMI along the Life-Cycle: Evidence from several US Cohorts

PLOS ONE

Dear Dr. Sanz-de-Galdeano,

Thank you for submitting your manuscript to PLOS ONE. After careful consideration, we feel that it has merit but does not fully meet PLOS ONE’s publication criteria as it currently stands. Therefore, we invite you to submit a revised version of the manuscript that addresses the points raised during the review process.

More specifically, you must add a 'strengths and limitations' section in the discussion section of your manuscript.

We look forward to receiving your revised manuscript.

Kind regards,

David Meyre

Academic Editor

PLOS ONE

Additional Editor Comments (if provided):

Please add a 'strengths and limitations' section in the discussion section.

---

## [Author Response · Author response to Decision Letter 1]

28 Jul 2020

Ref: PONE-D-20-03465

Association of a Genetic Risk Score with BMI along the Life-Cycle: Evidence from several US Cohorts

PLOS ONE

July 26th, 2020

Dear Referees, 

Please find enclosed a revised version of the paper, Manuscript Ref PONE-D-20-03465, “Association of a Genetic Risk Score with BMI along the Life-Cycle: Evidence from several US Cohorts”. We thank you and the Editor for your comments, and for giving us the opportunity to revise the paper. We believe the paper has considerably improved as a result. Further details are in our point-to-point reply letters that follow.

Yours sincerely,

The authors

Thank you for submitting your manuscript to PLOS ONE. After careful consideration, we feel that it has merit but does not fully meet PLOS ONE’s publication criteria as it currently stands. Therefore, we invite you to submit a revised version of the manuscript that addresses the points raised during the review process.

More specifically, you must add a 'strengths and limitations' section in the discussion section of your manuscript.

We have added a section “strengths and limitations” in page 14, which includes the following points: 

1. In this paper, we use two longitudinal surveys to provide new evidence on gene-age interaction effects on BMI for several cohorts. In particular, we study teenagers from the Add Health cohort (born 1974-1983) as they transition into adulthood as well as individuals aged 55+ who were born in earlier HRS cohorts (1931-53) as they move into old-age. The use of panel data is crucial in this context because it allows one to disentangle age/time associations from cohort effects. In contrast, as argued by (Lasky-Su et. al., 2008), cross-sectional studies may fail to detect age-varying associations as they cannot disentangle age/time from cohort effects. 

Our analyses are based on different cohorts observed at different stages of the life cycle. Hence, our contrasting findings for Add Health and the HRS may reflect differing patterns of genetic influence along the life cycle, but they could also stem from systematic differences across cohorts in their life-cycle patterns of genetic influence.

2. Note also that in this paper we estimate a reduced-form model without digging into the mechanisms behind gene-age interactions because of data limitations. Our results therefore could be explained by changes in the biology of BMI across the life course as well as by environmental changes that may reinforce or mitigate the effect of genes on BMI (Sanz‐de‐Galdeano and Terskaya, 2019; Liu et. al. 2015; Barcellos et. al, 2018). Understanding the mechanisms behind the patterns we uncover is worth further investigation. 

3. Another limitation of our analyses is that the genome-wide association study employed to compute the BMIPGS used mostly relies on European-descent individuals (Locke at. al., 2015). Therefore, our results cannot be generalized to individuals from different ancestries. The availability of GWAS for other ancestries would allow to overcome this limitation.

4. Finally, another potential limitation stems from the fact that the strength of genotype-phenotype associations may vary by age. Hence, GWAS results may not replicate in samples where the age distribution differs from that of the GWAS sample (Lasky-Su et. al., 2008). The BMIPGS we use rely on the GWAS conducted by Locke at. al. (2015). which is in turn mostly based on a sample of midlife individuals. Hence, their predictive power may be lower for younger individuals. A similar argument may apply to other demographic characteristics like childhood socioeconomic status (as our Add Health results by socioeconomic status suggest). While the strongest BMIPGS-BMI association we uncover is for young adults (Waves 4 and 5 of Add Health), this warrants further investigation. 

Barcellos SH, Carvalho LS, Turley P. Education can reduce health differences related to genetic risk of obesity. Proceedings of the National Academy of Sciences. 2018;115(42):E9765-E9772.

Lasky-Su J, Lyon HN, Emilsson V, Heid IM, Molony C, Raby BA, et al. On the replication of genetic associations: timing can be everything! The American Journal of Human Genetics. 2008;82(4):849-858.

Liu H, Guo G. Lifetime socioeconomic status, historical context, and genetic inheritance in shaping body mass in middle and late adulthood. American sociological review. 2015;80(4):705-737.

Locke AE, Kahali B, Berndt SI, Justice AE, Pers TH, Day FR, et al. Genetic studies of body mass index yield new insights for obesity biology. Nature. 2015;518(7538):197.

Sanz-de Galdeano A, Terskaya A. Sibling Differences in Educational Polygenic Scores: How Do Parents React? IZA Discussion Paper No. 12375; 2019

---

## [Editor Report · Decision Letter 2]

31 Aug 2020

Association of a Genetic Risk Score with BMI along the Life-Cycle: Evidence from several US Cohorts

PONE-D-20-03465R2

Dear Dr. Sanz-de-Galdeano,

We’re pleased to inform you that your manuscript has been judged scientifically suitable for publication and will be formally accepted for publication once it meets all outstanding technical requirements.

Kind regards,

David Meyre

Academic Editor

PLOS ONE
---

## [Editor Report · Acceptance letter]

4 Sep 2020

PONE-D-20-03465R2 

Association of a Genetic Risk Score with BMI along the Life-Cycle: Evidence from several US Cohorts  

Dear Dr. Sanz-de-Galdeano:

I'm pleased to inform you that your manuscript has been deemed suitable for publication in PLOS ONE. Congratulations! Your manuscript is now with our production department. 

Kind regards, 

on behalf of

Dr. David Meyre 

Academic Editor

PLOS ONE